

# Berberine protects hepatocyte from hypoxia/reoxygenation-induced injury through inhibiting circDNTTIP2

Yi Zhu[1,2], Junhui Li[1,2], Pengpeng Zhang[1,2], Bo Peng[1,2], Cai Li[1,2], Yingzi Ming[1,2] and Hong Liu[1,2]

[1] The Third Xiangya Hospital, Central South University, Changsha, China
[2] Engineering and Technology Research Center for Transplantation Medicine of National Health Commission, Changsha, China

Corresponding author
Hong Liu, 601941@csu.edu.cn

## ABSTRACT

**Background:** During hepatic ischemia-reperfusion injury, the excessive release of inflammatory cytokines can activate the intracellular signal transduction cascade to induce hepatocyte injury. Apoptosis is an important way of cell death after I/R injury. Berberine, a common quaternary ammonium alkaloid, has anti-inflammatory, anti-oxidative stress, and anti-apoptotic effects. An increasing number of studies have revealed the importance of non-coding RNAs, including microRNA, long non-coding RNAs and circular RNAs (circRNAs), as regulators of the effects of berberine.

**Purpose:** In this study, we investigated the mechanism of berberine against liver ischemia-reperfusion injury *in vitro*.

**Study Design and Methods:** In this study, hypoxia-reoxygenation (H/R)-treated L02 cells were pretreated with berberine to study the role and mechanism of berberine in resisting hepatic ischemia-reperfusion injury.

**Results:** The results show that berberine pre-treatment increased the cell viability of H/R-challenged cells, reduced H/R-induced apoptosis and ROS production, reversed H/R-increased on IL-6, IL-1β, TNF-α, and H/R-decreased IL-10 expression. Mechanically, berberine protect hepatocyte from H/R injury, at least partially, through circDNTTIP2. In addition, circDNTTIP2 can bind to the TATA box of caspase3 promoter, thereby promoting caspase 3-related cell apoptosis and the release of inflammatory cytokines.

**Conclusion:** This study found that berberine has a protective effect on H/R-induced hepatocyte damage by inhibiting a novel circRNA, circDNTTIP2. This study provides potential treatment strategies and treatment targets for liver ischemia-reperfusion injury.

## INTRODUCTION

Hepatic ischemia reperfusion (I/R) injury manifests in diverse clinical scenarios, such as liver surgery, transplantation, and cancer-related procedures (*Peralta, Jimenez-Castro & Gracia-Sancho, 2013*). The liver may initially manifest as direct cell damage caused by

ischemic injury, and reperfusion may further spread dysfunction and damage by activating inflammatory signaling pathways (*Lu et al., 2018*). Inflammatory cytokines are excessively released during I/R injury, triggering intracellular signaling cascades that contribute to hepatocyte damage (*Siriussawakul, Zaky & Lang, 2010*). Inflammation can lead to the secretion of cytokines/chemokines, which is associated with increased liver cell death during I/R (*Brenner et al., 2013*). Studies have unequivocally established apoptosis as a critical mechanism of cell death following I/R injury (*Jaeschke & Lemasters, 2003*). Therefore, inhibiting hepatocyte apoptosis caused by I/R is of great significance for hepatic I/R injury therapy. As of now, there exists no efficacious approach to prevent and alleviate hepatic I/R injury (*Galaris, Barbouti & Korantzopoulos, 2006*; *Jaeschke, 2003*). Consequently, it is imperative to explicate the mechanisms underlying the progression of hepatic I/R injury.

Berberine, a prevalent quaternary ammonium alkaloid, exhibits notable antibacterial properties and substantial therapeutic efficacy in addressing digestive system disorders, as evidenced by various studies (*Wang et al., 2017*; *Kumar et al., 2015*). Recent investigations have revealed that berberine possesses anti-inflammatory and anti-oxidative stress properties, exerting regulatory influence over immune responses and anti-apoptotic processes (*Daniel et al., 2017*; *Liu et al., 2019*; *McCubrey et al., 2017*).

An expanding body of research has underscored the significance of non-coding RNAs, encompassing microRNAs and lncRNAs, in modulating the anti-cancer and anti-diabetic effects attributed to berberine (*Ayati et al., 2017*; *Chang, 2017*). Circular RNAs (circRNAs), as a novel class of endogenous non-coding RNA, play pivotal roles as regulators of various cellular processes (*Chen & Shan, 2021*; *Li et al., 2022*; *Meng et al., 2023*; *Mugoni et al., 2022*; *Cai et al., 2023*; *Liao et al., 2023*; *Xia et al., 2021*; *Lin et al., 2023*; *Zhou et al., 2021*; *Gao et al., 2023*). The circRNA-miRNA-mRNA network represents a novel regulatory mechanism facilitating the effective treatment of diverse diseases using traditional Chinese medicine (*Zhang et al., 2020*). Nevertheless, the precise mechanism underlying the protective effects of berberine against liver ischemia-reperfusion injury remains elusive. In our investigation, we identified berberine as a potential mediator in ameliorating liver cell hypoxia reoxygenation (H/R) injury by suppressing circDNTTIP2. This inhibition resulted in the mitigation of Caspase 3-associated cell apoptosis, along with a reduction in cellular oxidative stress and the release of inflammatory factors. This study is expected to provide some novel insights into the treatment of hepatic ischemia reperfusion (I/R) injury.

## MATERIALS AND METHODS

### Cell culture

Human normal liver cells, namely L02 cells, were obtained from the Chinese Academy of Sciences Cell Bank (Shanghai, China). L02 is a commonly used human normal liver cell line and could be easily obtained commercially. The cells were cultured in a 37 °C, 5% carbon dioxide incubator using DMEM medium (Gibco, Carlsbad, CA, USA) supplemented with 10% fetal bovine serum (FBS; Gibco, Billings, MT, USA). The culture medium was refreshed every two to three times a week. When the cells reached

approximately 80% confluence, they were detached using trypsin and subcultured at a 1:4 ratio for subsequent passages.

## Establishment of hypoxia-reoxygenation cell model

L02 cells were seeded in a 6-well plate at a density of $1 \times 10^6$ cells per well and allowed to incubate (Forma3111; Thermo, Waltham, MA, USA) for 24 h. Following this, the cells were exposed to an anoxic environment (37 °C, 1% $O_2$, 5% $CO_2$, and 94% $N_2$) for 6 h. Subsequently, they were transferred to a regular incubator set at 37 °C, 5% $CO_2$, and 95% air for a 6-h reoxygenation period to facilitate the subsequent experimental procedures.

## Berberine pretreatment of liver cells

The L02 cells were divided into the following groups: a normal control group without H/R treatment, an H/R treatment group, and various concentrations of berberine (Sigma, St. Louis, MO, USA) pretreatment groups (0, 2, 4, 8, 16 μM berberine pretreatment for 24 h) to determine the optimal drug dose. The H/R treatment consisted of a 6-h period of hypoxia followed by 6 h of reoxygenation.

## Cell transfection

The synthesized circDNTTIP2 expression plasmid, circDNTTIP2 negative control (negative control), circDNTTIP2 siRNA (sequence: TTTCCGATTTGATCAAACGACGATTA), and scramble control were provided by Guangzhou Ruibo Biological Co., Ltd (Guangzhou, China). For transfection, L02 cells were seeded in a six-well plate at a density of $1 \times 10^6$ cells per well and allowed to reach 70% confluence. Transfection was performed using Lipofectamine 2000 (Invitrogen, Carlsbad, CA, USA) according to the provided instructions.

## Cell viability assay

Cell viability was evaluated using the CCK-8 method. After transfection and exposure to H/R, L02 cells were seeded in a 96-well plate at a density of $7 \times 10^3$ cells per well. Next, 10 μL of CCK-8 reagent (C0038; Beyotime, Nantong, China) was added to each well, and the plate was incubated in the dark for 3 h. The optical density (OD) values at a wavelength of 450 nm were measured using an automatic microplate reader from Thermo Fisher Company (Waltham, MA, USA). To determine cell viability, the readings were normalized to the scramble treatment group.

## Flow cytometry to detect apoptosis

Cell apoptosis was assessed using the Annexin V-FITC apoptosis kit (Beyotime, Nantong, China) following the provided instructions. Briefly, the cells were suspended in 195 μL of Annexin V-FITC binding solution. Then, 5 μL of Annexin V-FITC and 10 μL of propidium iodide (PI) were added to the suspension and gently mixed. After incubating the mixture at room temperature in the dark for 20 min, flow cytometry (CytoFLEX S; Beckman, Brea, CA, USA) was employed to analyze the apoptotic cells. Data processing and analysis were performed using CytExpert/FlowJo software.

## Reactive oxygen species measurement

The quantification of intracellular reactive oxygen species (ROS) levels was performed using the ROS Assay Kit (Beyotime, Shanghai, China), which relies on the fluorescent dye DCFH-DA to measure changes in fluorescence intensity. Following the respective treatments, L02 cells were suspended in a diluted solution of DCFH-DA. The solution was prepared by diluting DCFH-DA with serum-free culture medium (Gibco, Carlsbad, CA, USA) at a 1:1,000 ratio, resulting in a final concentration of 10 µmol/L. The cell density ranged between $1.0 \times 10^6$ and $2.0 \times 10^7$ cells. Incubation of the cells at 37 °C in a cell incubator for 20 min allowed for DCFH-DA uptake. Afterward, the cells were thoroughly washed three times with serum-free cell culture medium to remove any unabsorbed DCFH-DA. Finally, fluorescence intensity was directly observed using a confocal laser microscope (TS-100; Nikon, Tokyo, Japan), with excitation at a wavelength of 488 nm and emission detection at 525 nm.

## CircRNA sequencing

CircRNA sequencing was conducted following established protocols and procedures, as previously described (*Wu et al., 2021*). The Arraystar Human circRNA Microarray, capable of detecting approximately 5,816 circRNAs, was employed for comprehensive analysis of human circRNA expression. Sample labeling and chip hybridization followed the Agilent One Color Microarray Based Gene Expression Analysis protocol. mRNA isolation was performed on H/R-exposed L02 cells, with untreated L02 cells as the negative control, using the mRNA ONLY Eukaryotic mRNA Isolation Kit (Qiagen, Hilden, Germany). Enrichment and transcription of each sample into fluorescent cRNA were carried out using random primers. The labeled cRNA was then purified, and its concentration and activity were measured with the NanoDrop ND-1000 (Thermo Fisher Scientific, Waltham, MA, USA). Subsequently, the labeled cRNA was combined with blocking solution, lysis buffer, and hybridization buffer. The hybridization mixture was incubated on a glass slide in an Agilent hybrid incubator for 17 h. Following the cleaning, fixing, and scanning of the hybridization chip with the Agilent DNA Microarray Scanner, chip image analysis was performed using Agilent Feature Extraction software. Quantile standardization and data processing were conducted with the Greenspring GX software package. Further data analysis focused on selecting mRNAs with defined values, and Agilent Greenspring GX software was utilized to generate a volcano map.

## RNA quantitative real-time polymerase chain reaction

Total RNA was extracted using a TRIzol kit (Invitrogen, Waltham, MA, USA) and cDNA synthesis was performed with a corresponding reverse transcription kit (Qiagen, Hilden, Germany). Real-time polymerase chain reaction (RT-PCR) was carried out on an ABI7900HT instrument (Applied Biosystems, Foster City, CA, USA) using a SYBR Green SuperMix kit (Bio-Rad, Hercules, CA, USA). The RT-PCR reaction conditions included an initial denaturation step at 95 °C for 10 s, followed by 40 cycles of amplification consisting of 95 °C for 5 s, 60 °C for 30 s, and 72 °C for 30 s. The primer sequences are listed in Table 1.

**Table 1 Primers.**

| Primer name | qPCR primer sequence (5′→3′) |
| --- | --- |
| hsa_circ_0005199-Forward | AGTCTGACAGCCACA TCACA |
| hsa_circ_0005199-Reverse | TGACTCCTTCACTGCTCTGG |
| hsa_circ_0110306-Forward | TGAGGCTTCAGAAATTGTCCAG |
| hsa_circ_0110306-Reverse | CCAGCTTCCCATCTTTGTTGA |
| hsa_circ_0000117-Forward | TAGGACATATGGGTGGGGAC |
| hsa_circ_0000117-Reverse | GCCTTCTCATGATCAGCTCG |
| hsa_circ_0008339-Forward | CTGTCTATCACCAGCCAGCA |
| hsa_circ_0008339-Reverse | TGGCAGCTTTTACCACTCCA |
| hsa_circ_0111350-Forward | ACAACAGTCAGACTCAGCCA |
| hsa_circ_0111350-Reverse | AGCATCACCTCCTTTTCCCA |
| hsa_circ_0016601-Forward | GCAGTTACAGCAAGCTACCC |
| hsa_circ_0016601-Reverse | TCTTGGTTTCCTCCTTGTCCA |
| hsa_circ_0013218-Forward | TGGAAGAGGAAGACAAGGCA |
| hsa_circ_0013218-Reverse | TGCGCTTGAATCCCATTAGC |
| hsa_circ_0112397-Forward | GCTGCTCTTTCGAATCGTCA |
| hsa_circ_0112397-Reverse | ACATCTGGCTGGTCATGAGT |
| hsa_circ_0012152-Forward | ATCGGGAAGCCTCATCTAGC |
| hsa_circ_0012152-Reverse | CAGAGGGTTGGGAAGGTAGG |
| hsa_circ_0002563-Forward | AATCAACCCTGCTACCGGAA |
| hsa_circ_0002563-Reverse | TGTCTTGCCACTTCCTGTCT |
| DNTTIP2-Forward | GGAAGAGGGAAGTCGTGGTG |
| DNTTIP2-Reverse | GTCCCCTTTGGTAGTGAGCC |
| IL-1β-Forward | TGAGCTCGCCAGTGAAATGA |
| IL-1β-Reverse | AACACGCAGGACAGGTACAG |
| IL-6-Forward | GTCCAGTTGCCTTCTCCCTGG |
| IL-6-Reverse | CCCATGCTACATTTGCCGAAG |
| IL-10-Forward | TTGCAAAAGAAGGCATGCACAG |
| IL-10-Reverse | ATAGAGTCGCCACCCTGATG |
| TNFα-Forward | CACAGTGAAGTGCTGGCAAC |
| TNFα-Reverse | ACATTGGGTCCCCCAGGATA |
| Caspase 3-Forward | GCTCATACCTGTGGCTGTGT |
| Caspase 3-Reverse | TCTGTTGCCACCTTTCGGTT |
| β-actin-Forward | GGGAAATCGTGCGTGACATTAAG |
| β-actin-Reverse | TGTGTTGGCGTACAGGTCTTTG |

## RNase R treatment

The L02 cells subjected to H/R treatment were seeded in a six-well plate at a density of $1 \times 10^6$ cells per well. After 24 h of incubation, the cells achieved a confluence of 70%. For qPCR analysis, 2 μg of total RNA was combined with 3 U/μg RNase R from Epicentre Technologies (Madison, WI, USA) and incubated at 37 °C for 15 min.

### RNA binding protein immunoprecipitation

RIP was performed as previously described with some modifications (*Wu et al., 2021*). After a 48-h transfection period, cells were collected and subjected to lysis using RIP lysis buffer (Sigma, St. Louis, MO, USA) on ice for 10 min. Following centrifugation, the resulting supernatant was combined with 30 µl of Protein-A/G magnetic beads (Roche, Pleasanton, CA, USA) and 10 µg of caspase 3 antibody (Abcam, Boston, MA, USA) and incubated overnight at 4 °C. Subsequently, the immunocomplex was centrifuged, and the obtained pellet was subjected to three washes using washing buffer. The eluted RNA from this process was utilized for qRT-PCR analysis.

### Dual-Luciferase reporter gene assay

The Dual-Luciferase reporter gene assay, based on the methods described by *Wu et al. (2021)* with modifications, was conducted. A recombinant vector, synthesized by Songon Biotech Corporation (Shanghai, China), containing wild-type (WT) or mutant (Mut) Caspase 3 promoter fragments corresponding to the predicted target binding site of circDNTTIP2, was used. L02 cells were transfected in a 24-well culture plate with circDNTTIP2 (800 ng) or circDNTTIP2 siRNA (20 pmol), along with either the wild-type or mutant Caspase 3 promoter (20 pmol), using Lipofectamine 2000 following the manufacturer's instructions. After 48 h of transfection, cell harvesting was performed, and luciferase activity was measured using the Thermo Scientific Vanquish system (Thermo Fisher Scientific, Waltham, MA, USA).

### Statistical analysis

Statistical analysis was performed using SPSS 16.0 software. The experimental data, obtained from three independent repetitions, are presented as mean ± SD. To assess statistical significance ($P < 0.05$), one-way analysis of variance (ANOVA) was conducted, followed by Bonferroni's *post-hoc* test.

## RESULTS

### Berberine's protective effect against H/R-induced injury in L02 cells

We investigated the protective effect of berberine against H/R-induced injury by evaluating cell viability, apoptosis, and ROS production in berberine pre-treated L02 cells exposed to H/R. After a 24-h pre-treatment with berberine, the cells were subjected to 6 h of hypoxia followed by 6 h of reoxygenation. H/R significantly reduced cell viability compared to the control group. However, berberine pre-treatment at concentrations of 2, 4, 8, and 16 µM increased cell viability, with the most significant effect observed at 8 µM (Fig. 1A). Therefore, we selected the 8 µM concentration for further analysis. Moreover, berberine pre-treatment reduced the occurrence of H/R-induced apoptotic cells and ROS production (Figs. 1B, 1C). Additionally, berberine pre-treatment effectively reversed the H/R-induced elevation in IL-6, IL-1β, and TNF-α levels, while restoring the decreased IL-10 expression. These results demonstrate that berberine pre-treatment protects against H/R-induced L02 cell death and regulates cytokine levels.

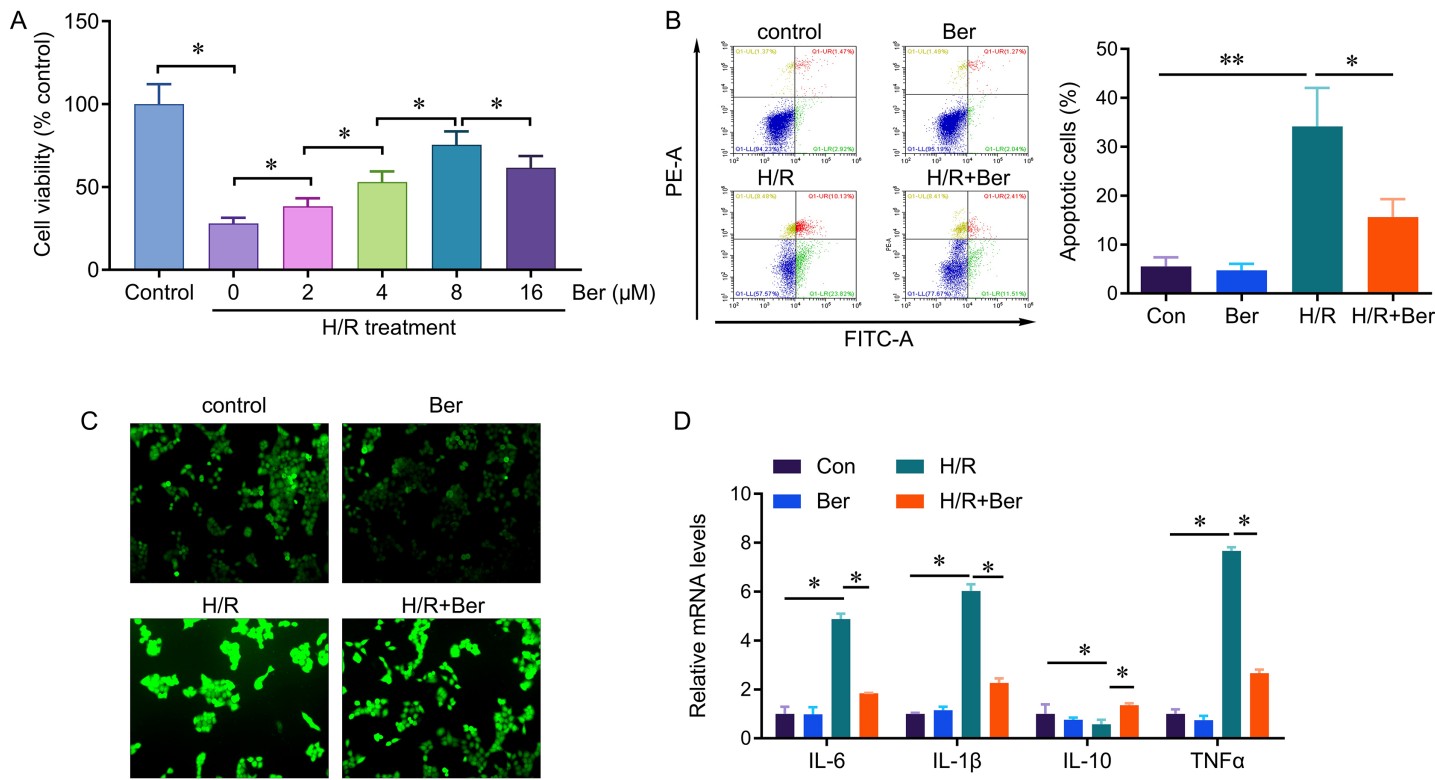

**Figure 1 Protective effect of berberine against H/R-induced injury on L02 cells.** *P < 0.05; **P < 0.01.

## Berberine protect hepatocytes from H/R injury through circDNTTIP2

To investigate the underlying mechanism of berberine's protective effect against H/R injury in hepatocytes, we performed circRNA sequencing to profile circRNA expression in H/R-treated L02 cells. Differential expression of circRNAs was observed between H/R-treated and control cells, with 1,078 circRNAs upregulated and 1,235 circRNAs downregulated in the H/R-treated cells (Fig. 2A). To validate these findings, we conducted qPCR analysis on 10 upregulated circRNAs. The results confirmed the upregulation of circ_0000117, circ_0008339, circ_0111350, circ_0016601, circ_0013218, circ_0112397, and circ_0012152 upon H/R treatment, while circ_0005199, circ_0110306, and circ_0002563 remained unchanged (Fig. 2B). Notably, circ_0013218 showed a significant over 15-fold upregulation and was designated as circDNTTIP2 due to its origin from exon 2 of the DNTTIP2 gene (Fig. 2B). The circular structure of circDNTTIP2 was further confirmed through RNase R-resistant exonuclease digestion (Fig. 2C). Interestingly, both H/R and berberine treatment did not influence the expression of DNTTIP2 mRNA. However, berberine pre-treatment attenuated the H/R-induced upregulation of circDNTTIP2 (Fig. 2D), prompting us to select circDNTTIP2 for further investigation.

Subsequently, we introduced circDNTTIP2 overexpression in L02 cells (Fig. 3A) and observed that its overexpression significantly exacerbated H/R-induced inhibition of cell proliferation and apoptosis while diminishing the protective effects of berberine pre-treatment on cell survival (Figs. 3B, 3C). Moreover, circDNTTIP2 overexpression
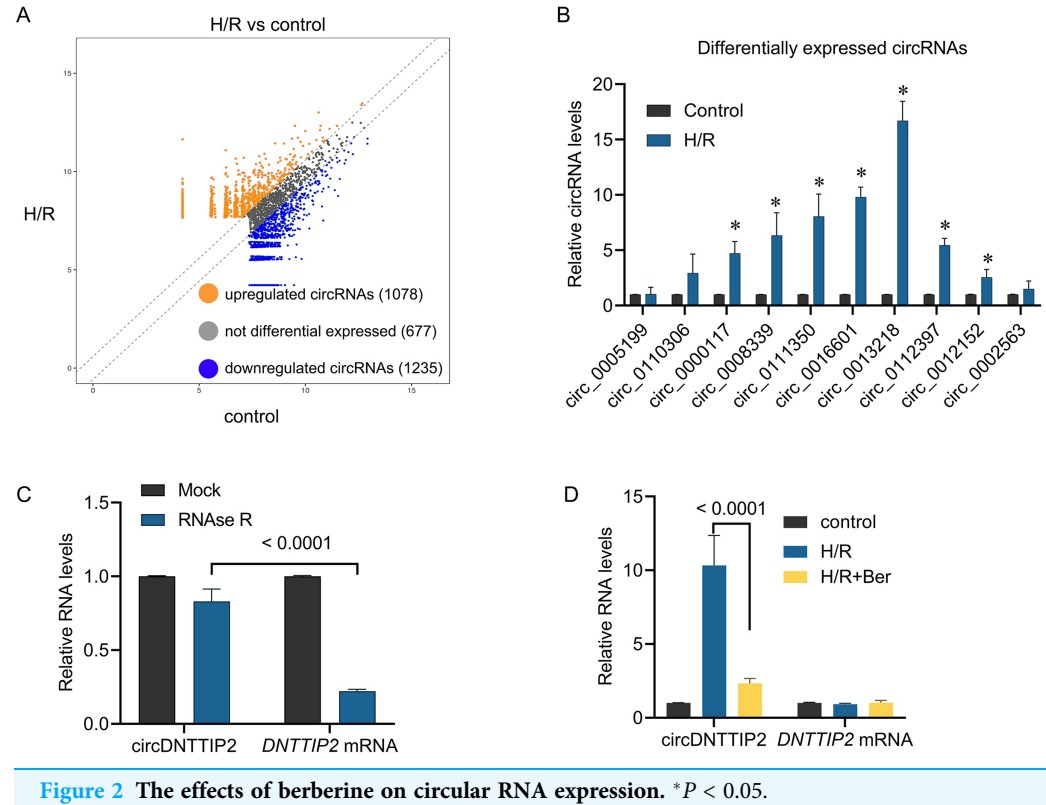

**Figure 2** The effects of berberine on circular RNA expression. *$P < 0.05$.

amplified the H/R-induced expression of IL-6, IL-1β, and TNF-α, while reversing the effects of berberine pre-treatment on these genes (Fig. 3D). Additionally, circDNTTIP2 overexpression increased ROS production compared to H/R treatment alone and counteracted the effects of berberine pre-treatment (Fig. 3E). These findings suggest that berberine protects hepatocytes from H/R injury, partially by inhibiting circDNTTIP2.

## circDNTTIP2 activates caspase 3 gene transcription

The involvement of caspase 3 in promoting cell apoptosis and releasing inflammatory factors during ischemia-reperfusion injury has been established in previous studies. In our investigation, we observed that overexpression of circDNTTIP2 resulted in an increased release of IL1β and IL-18. To investigate the regulatory role of circDNTTIP2 on caspase 3 expression, we assessed its impact on caspase 3 levels. Our findings confirmed that circDNTTIP2 overexpression enhanced caspase 3 expression, whereas knockdown of circDNTTIP2 reduced its expression (Fig. 4A). Subsequent analysis using the AREsite tool and BLAST analysis revealed a potential interaction between the AUAU motif of circDNTTIP2 and the TATA box in the caspase 3 promoter. To validate this interaction, we generated wild-type (caspase3-WT) and mutant (caspase3-Mut) constructs of the caspase 3 promoter. The overexpression of circDNTTIP2 significantly augmented the activity of the wild-type caspase3 promoter luciferase reporter gene, while it had no effect on the mutant caspase 3 promoter luciferase reporter gene (Fig. 4B). Similarly, knockdown of circDNTTIP2 markedly inhibited luciferase activity driven by the wild-type caspase3

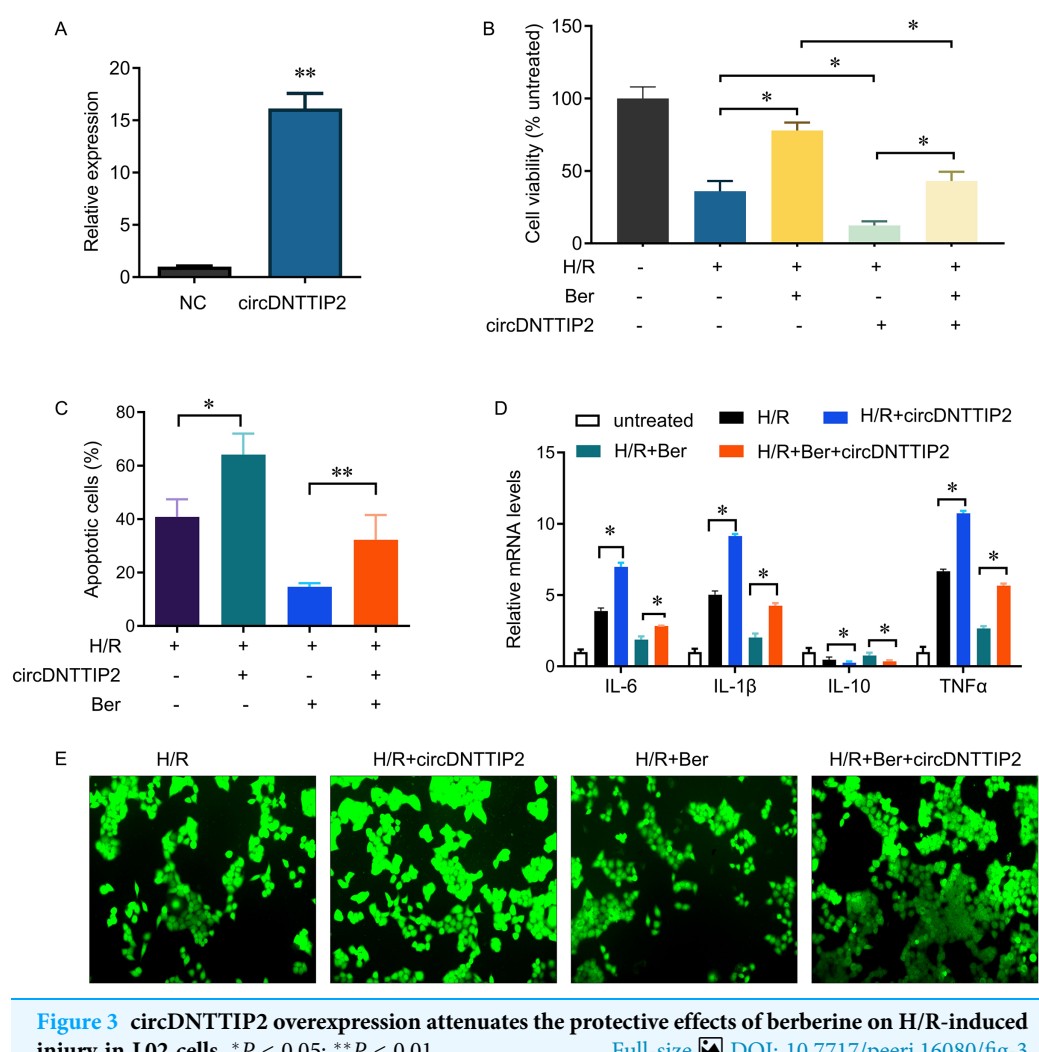

**Figure 3** **circDNTTIP2 overexpression attenuates the protective effects of berberine on H/R-induced injury in L02 cells.** $*P < 0.05$; $**P < 0.01$.

promoter, but not the mutant caspase3 promoter (Fig. 4C). Furthermore, RNA pull-down analysis substantiated the physical interaction between circDNTTIP2 and caspase3 (Fig. 4D).

## CircDNTTIP2 promotes H/R damage through caspase3

To investigate the relationship between circDNTTIP2 and the caspase3 pathway in liver hypoxia-reoxygenation (H/R) injury, we conducted further experiments. Our results demonstrated that depletion of circDNTTIP2 significantly mitigated the inhibitory effects of H/R on cell viability, ROS production, and cell apoptosis (Figs. 5A–5C). Remarkably, these effects were reversed by caspase3 overexpression. Furthermore, elevated caspase3 levels prominently enhanced the release of IL1β, IL6, and TNFα, which had been suppressed by circDNTTIP2 knockdown (Fig. 5D). Collectively, these findings provide strong evidence that circDNTTIP2 exacerbates H/R injury by potentiating caspase3 activity.

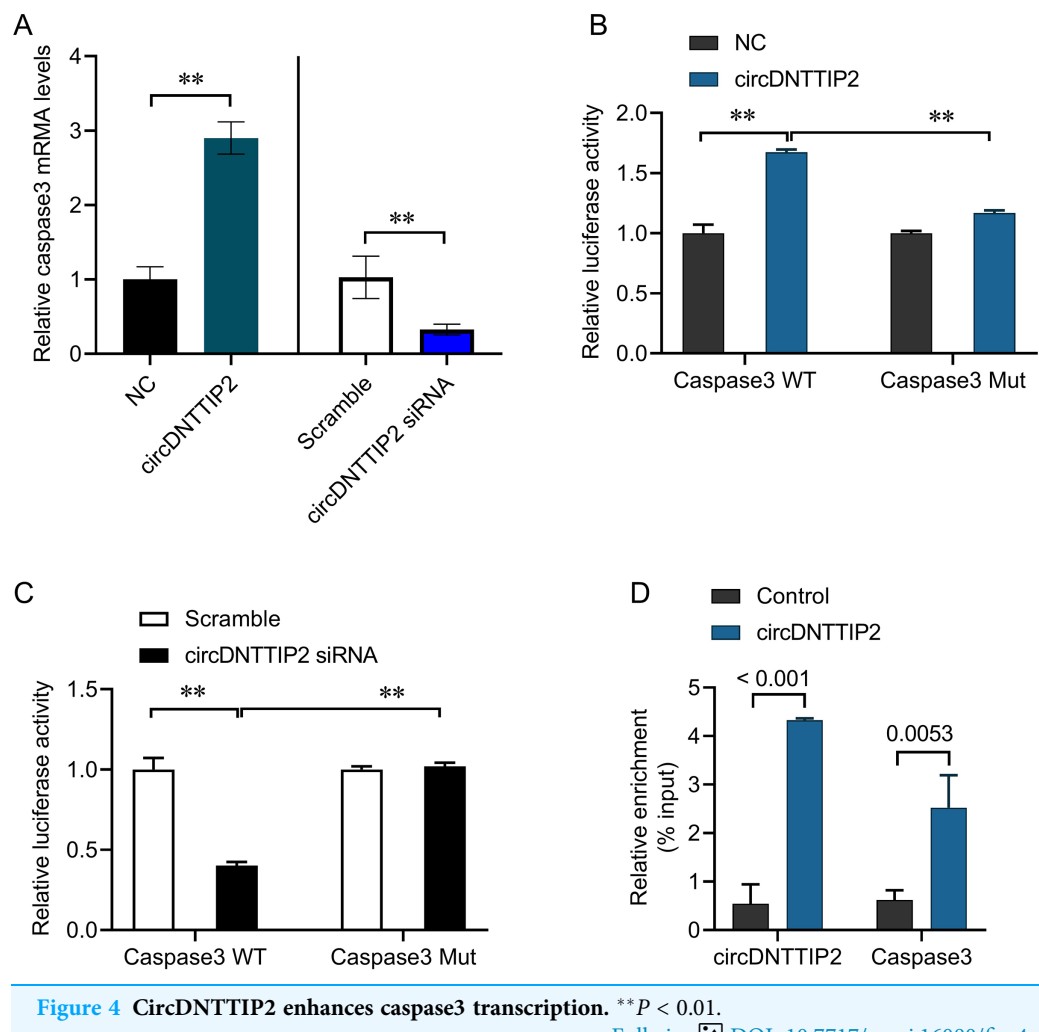

**Figure 4 CircDNTTIP2 enhances caspase3 transcription.** $^{**}P < 0.01$.

## DISCUSSION

Berberine is an isoquinoline alkaloid (molecular formula: $C_{20}H_{18}NO_4$), and is soluble in hot water and ethanol. Its hydrochlorideis were commonly used in clinic (*Yin, Zhang & Ye, 2008*). It has recently been reported in the literature that berberine increases the stability of LDLR mRNA by reducing the binding of the 3′ untranslated region of LDLR mRNA to heterogeneous ribonucleoprotein (*Li et al., 2009*). Berberine treatment can increase the phosphorylation levels of AMPK and ACC in the liver of db/db mice (*Lee et al., 2006*). Oral berberine can reduce blood LDL-C and TG levels in patients with liver disease (*Kong et al., 2004*; *Zhao et al., 2008*). In addition, berberine can inhibit the activation of NF-κB caused by hyperglycemia and the increase of pro-inflammatory factors and chemokines such as TNFα, IL1-β, IL8 and MCP1 (*Wang et al., 2009*). Studies have shown that berberine is closely related to cell damage and apoptosis caused by ischemia-reperfusion. For example, berberine pretreatment can reduce the area of cerebral infarction caused by focal cerebral ischemia/reperfusion. Further studies have confirmed that berberine significantly reduced the level of malondialdehyde in hippocampus and cortical neurons and increased the

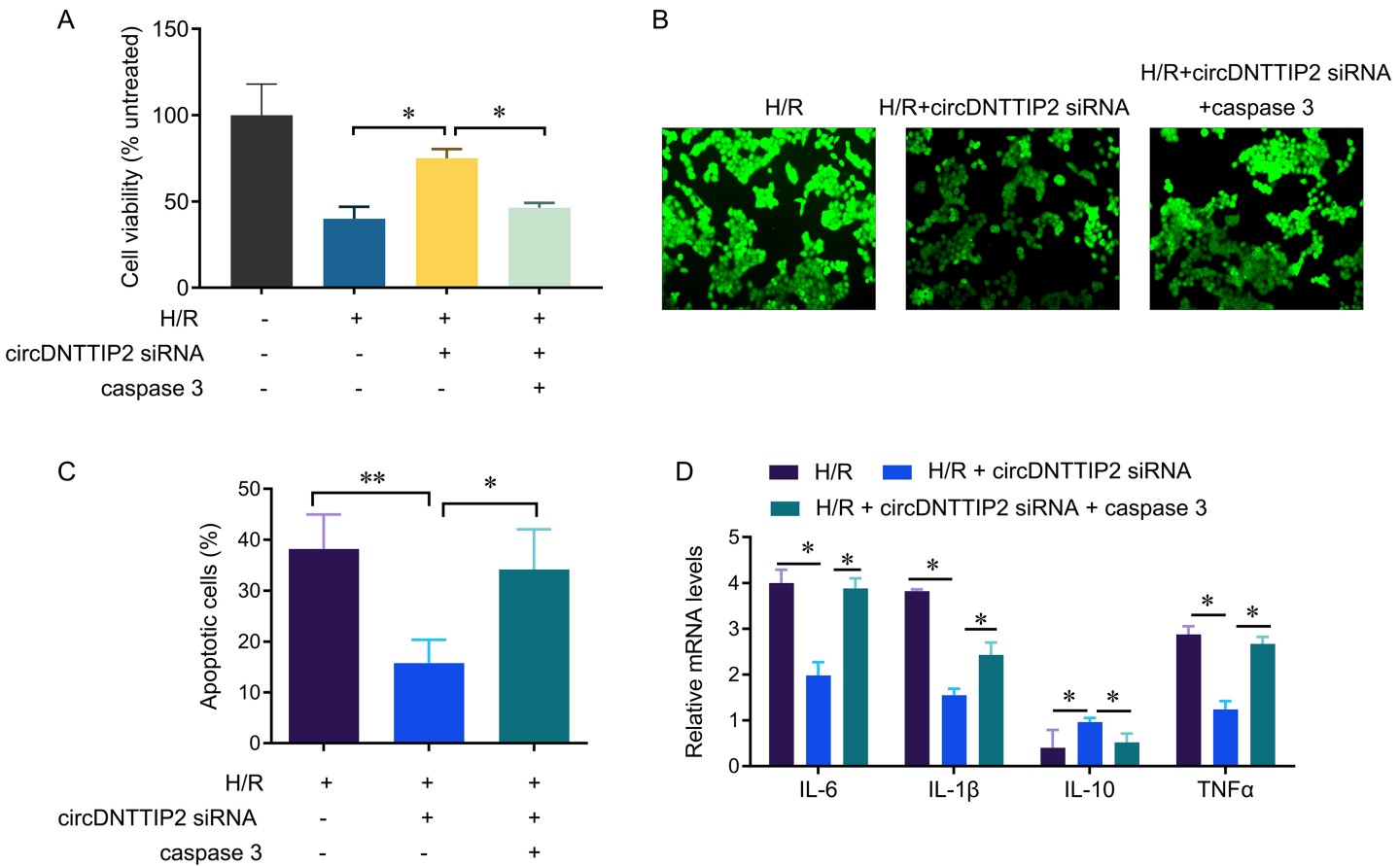

**Figure 5 CircDNTTIP2 exacerbates H/R injury through caspase3 pathway in L02 cells.** $^*P < 0.05$; $^{**}P < 0.01$.

activity of superoxide dismutase after cerebral ischemia in rats, indicating that berberine can protect ischemic brain tissue by reducing ROS (*Zhou et al., 2008*). *Hu et al. (2012)* found that the role of berberine in protecting neurons may be achieved by up-regulating the expression of PI3K-specific regulatory subunit p55γ, thereby inhibiting the activation of the apoptotic proteins Bad and Caspase-3. These results suggest that berberine can inhibit the release of inflammatory factors and reduce ischemia/reperfusion injury.

At present, there are few reports on the effect of berberine on ischemia/reperfusion liver cell injury. This study found that berberine pretreatment in hepatocytes can inhibit H/R-induced ROS aggregation and release of inflammatory factors, and reduce cell death. Mechanism studies have found that berberine treatment reduced the expression of circDNTTIP2 induced by H/R, while circDNTTIP2 promoted cell apoptosis and the release of inflammatory factors by enhancing caspase 3. Studies have shown that liver ischemia-reperfusion injury, liver steatosis, hepatitis B, and liver regeneration are closely associated with circRNA regulatory disorders. For example, *Ye et al. (2018)* used bioinformatics analysis methods to screen out two up-regulated and three down-regulated circRNAs in liver tissues of liver IR model mice, and further analyzed by GO that these circRNAs mainly play a role in cellular components and molecular functions. At the same

time, through the KEGG method screening and analysis, it is concluded that the Hippo signal transduction pathway is the key signal transduction pathway that causes the significant differential expression of circRNA in the liver tissue of IR model mice, suggesting that the differentially expressed circRNA can be controlled by specific signal transduction pathways, further affecting the occurrence and development of liver IR (*Ye et al., 2018*). *Guo et al. (2017a*, *2017b)* found that circRNA_0046367 as an endogenous regulator of miR-34a could change the process of liver steatosis; circRNA_0046367 blocked the interaction between miRNA/mRNA and miRNA response elements and eliminate the inhibitive effect of miR-34a on perixisome proliferation-activated receptor alpha (PPARα), thereby preventing lipid peroxidation related to steatosis. *Zhou et al. (2018)* performed RNA sequencing analysis on the liver tissues of chronic hepatitis B patients and normal control groups, and screened out the differentially expressed hsa_circ_0000650; and found that the pathway of hsa_circ_0000650-miR-6873-3p-TGF-β2 has an inhibitory effect on IL-2-dependent T cell growth, suggesting that differentially expressed circRNA may participate in the occurrence of chronic hepatitis B through the circRNA-miRNA-mRNA pathway. CircRNAs were also associated with steroid hormone biosynthesis and inflammatory mediator regulation in transient receptor potential channels during the regeneration and repair process after liver injury (*Li et al., 2017*).

CircDNTTIP2 is a circular RNA molecule that has been extensively studied in recent years. It is derived from the DNTTIP2 gene and has been found to have various functions in different biological processes. CircDNTTIP2 has been shown to be involved in promoting hypoxia-reoxygenation damage through the caspase3 pathway. This pathway is a key player in programmed cell death, also known as apoptosis. When cells are exposed to hypoxia, which is a condition of low oxygen levels, and then reoxygenated, it can lead to cellular damage and ultimately cell death. CircDNTTIP2 has been found to enhance this process by activating the caspase3 pathway, which triggers a cascade of events that ultimately result in cell death. Understanding the role of circDNTTIP2 in hypoxia-reoxygenation damage can provide valuable insights into potential therapeutic targets for conditions related to oxygen deprivation and reoxygenation injury.

circDNTTIP2 has also been found to play a role in promoting hypoxia-reoxygenation damage through the caspase3 pathway. When cells are exposed to hypoxia, which is a condition of low oxygen levels, circDNTTIP2 expression is upregulated. This upregulation leads to an increase in caspase3 activity, which is a key enzyme involved in programmed cell death, also known as apoptosis. The activation of caspase3 triggers a cascade of events that ultimately leads to cell death. Therefore, it can be inferred that circDNTTIP2 promotes hypoxia-reoxygenation damage by enhancing caspase3-mediated apoptosis.

In conclusion, this study found that berberine has a protective effect on H/R-induced hepatocyte damage. We identified a new circRNA, circDNTTIP2. circDNTTIP2 promotes the expression of caspase 3 by binding to the TATA box of the caspase 3 promoter, thereby exacerbating H/R-induced hepatocyte apoptosis and the release of inflammatory factors. Hepatic ischemia-reperfusion injury is an extremely complex process. Although the current research on the relationship between circRNA and liver injury has been in-depth, the specific formation mechanism, biological function of circRNA and its role in liver

ischemia-reperfusion injury are still not very clear. An in-depth understanding of the role of circRNA in the process of liver ischemia-reperfusion injury can provide therapeutic targets for liver diseases and further improve the therapeutic effect of liver diseases. The lack of *in vivo* validation and the use of only one cell line (LO2) is the limitations of the study, which should be addressed in the following research.

## ABBREVIATIONS

| | |
|---|---|
| **circRNAs** | circular RNAs |
| **H/R** | Hypoxia-reoxygenation |
| **I/R** | Ischemia reperfusion |
| **RIP** | RNA binding protein immunoprecipitation |
| **RT-PCR** | Real-time polymerase chain reaction |

### Funding

This study was supported by grants from the Natural Science Foundation of Hunan Province (2021JJ40959). The funders had no role in study design, data collection and analysis, decision to publish, or preparation of the manuscript.

### Grant Disclosures

The following grant information was disclosed by the authors:
Natural Science Foundation of Hunan Province: 2021JJ40959.

### Competing Interests

The authors declare that they have no competing interests.

### Author Contributions

- Yi Zhu conceived and designed the experiments, performed the experiments, analyzed the data, prepared figures and/or tables, authored or reviewed drafts of the article, and approved the final draft.
- Junhui Li analyzed the data, prepared figures and/or tables, authored or reviewed drafts of the article, and approved the final draft.
- Pengpeng Zhang analyzed the data, prepared figures and/or tables, authored or reviewed drafts of the article, and approved the final draft.
- Bo Peng analyzed the data, prepared figures and/or tables, authored or reviewed drafts of the article, and approved the final draft.
- Cai Li analyzed the data, authored or reviewed drafts of the article, and approved the final draft.
- Yingzi Ming analyzed the data, authored or reviewed drafts of the article, and approved the final draft.

- Hong Liu conceived and designed the experiments, performed the experiments, analyzed the data, prepared figures and/or tables, authored or reviewed drafts of the article, and approved the final draft.

## Data Availability

The raw measurements are available in the Supplemental Files.

## Supplemental Information

Supplemental information for this article can be found online at http://dx.doi.org/10.7717/peerj.16080#supplemental-information.

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
