# Peer review of "Berberine protects hepatocyte from hypoxia/reoxygenation-induced injury through inhibiting circDNTTIP2"

_PeerJ, doi:10.7717/peerj.16080_

## Round 0.1 · original submission · Major Revisions

The authors should address these Reviewers' comments.

·

Basic reporting

Zhu et al. investigated the role of berberine, a common quaternary ammonium alkaloid, in hepatic ischemia-reperfusion injury.  Berberine has a protective effect on H/R-induced hepatocyte damage by inhibiting a novel circRNA, circDNTTIP2, resulting in reduction of caspase 3-related cell apoptosis and the release of inflammatory cytokines. The research team adopted in vitro models compared with control group, the mechanism research method is relatively comprehensive. This study is valuable for understanding the mechanism of berberine in IRI. In conclusion, berberine can alleviate IRI and suppress H/R-induced apoptosis, ROS production, and inflammation via circDNTTIP2 pathway. However, several minor problems in the manuscript need to be improved before accepted.

Experimental design

The manufacturer of reagent and instrument used in the study are better to be described in a uniform format.

Validity of the findings

It is recommended to perform western blot analysis of caspase-3 in Figure.4 to confirm the expression of caspase-3.

Additional comments

The language of the manuscript needs to be polished, making it clearer.

Reviewer 2 ·

Basic reporting

The results demonstrated that pretreatment with berberine improved cell viability, reduced apoptosis, and suppressed the production of reactive oxygen species (ROS) in H/R-challenged cells. Berberine also reversed the upregulation of inflammatory cytokines, including IL-6, IL-1β, and TNF-α, while restoring the decreased expression of the anti-inflammatory cytokine IL-10. The study further revealed that berberine's protective effect was partially mediated by a novel circular RNA, circDNTTIP2. CircDNTTIP2 interacted with the TATA box of the caspase3 promoter, triggering caspase 3-related cell apoptosis and the release of inflammatory cytokines. Overall, this research sheds light on the protective properties of berberine against hepatocyte damage induced by I/R injury. By targeting circDNTTIP2, berberine holds promise as a potential therapeutic strategy for liver ischemia-reperfusion injury.
I appreciate the efforts of the authors in conducting such a study. Although this study is well-designed, some concerns need to be addressed.

Experimental design

1. Highlight the rationale for selecting Berberine as the study object in the introduction section. Clearly explain the specific reasons why Berberine was chosen, such as its known properties or previous research indicating its potential therapeutic effects. This will provide readers with a better understanding of the motivation behind studying Berberine in the context of the research topic.

2. Enhance the logic and thoroughness of the discussion section. Ensure that all results are adequately explained and discussed in detail. Address any inconsistencies or gaps in the interpretation of the findings. Provide further insights into the implications of the results, their potential mechanisms, and their relevance to the broader field of research. This will strengthen the overall argumentation and provide a comprehensive understanding of the study outcomes.

3. Engage a professional editing service to improve the language and correct any typos in the main manuscript. It is essential to ensure that the manuscript is free from grammatical errors, spelling mistakes, and other language-related issues. A polished and well-edited manuscript will enhance the readability and professionalism of the research.

4. Highlight the limitations of the study in the conclusion section, including the lack of in vivo validation and the use of only one cell line (LO2). Discuss the potential impact of these limitations on the generalizability and robustness of the findings. Acknowledge the need for future research to address these limitations and provide suggestions for further investigation. This will demonstrate a comprehensive understanding of the study's scope and potential areas for improvement.

5. Provide a thorough discussion of the background of circDNTTIP2. Elaborate on its known functions, potential regulatory mechanisms, and relevance to the research topic. Discuss how the findings of the study contribute to the existing knowledge about circDNTTIP2 and its implications for the broader field. This will enrich the understanding of the specific molecular context and highlight the significance of the research.

6. Clearly state the reason for choosing the LO2 cell line for subsequent studies right from the beginning. Justify the selection by explaining its relevance to the research objectives, its characteristics, and any previous research supporting its use. This will help readers understand why the LO2 cell line was specifically chosen and establish its importance in the context of the study.

Validity of the findings

no comment

Reviewer 3 ·

Basic reporting

no comment

Experimental design

no comment

Validity of the findings

no comment

Additional comments

In this study, the authors demonstrate the protective effect of berberine on H/R-induced hepatocyte damage by inhibiting a novel circRNA, circDNTTIP2. The research suggests potential therapeutic approaches and injury targets for liver ischemia-reperfusion damage. The research is fascinating. However, I still have some recommendations for how to make the current version better.

1) Ensure that the method section contains enough information to allow for simple study replication. Include detailed explanations of all measurements, data collection techniques, experimental procedures, and any unique protocols used. This will make it possible for other researchers to precisely replicate the study and verify the findings.

2) In the introduction section, provide more details about the study's significance. Indicate specifically how the study advances the field, filling in any knowledge gaps or revealing new insights. Discuss the research's potential implications, including how clinical practice or future research directions may be affected. This will make the study's significance and effects clearer to readers.

3) Where necessary, add information about the statistical analysis to the figure legends. Include details about the statistical tests employed, the levels of significance, and any multiple comparisons adjustments. The results shown in the figures will be more transparent and reproducible as a result.

4) In either the introduction or the discussion section, emphasize the context of circDNTTIP2. Describe the system's known functions, regulatory mechanisms, and any findings from prior studies that support the study's relevance. Readers will gain a thorough understanding of the particular circRNA and its significance within the larger research context as a result.

5) Check that the word "Berberine" is consistently spelt throughout the entire manuscript. Verify the spelling in every section, including the tables, figure legends, and the main text. Terminology consistency is essential for professionalism and clarity.

6) In the results section, under the heading "CircDNTTIP2 promotes H/R damage through caspase3," give a thorough explanation of the function of caspase3 in the discussion section that follows. Discuss its potential involvement in the cellular pathways impacted by circDNTTIP2, its relevance to the observed results, and its implications for the study's findings. This will make it easier to comprehend caspase3's function within the context of the study.

7) Check the sentence descriptions to make sure they don't read like Wikipedia articles. Verify the manuscript's accuracy, clarity, and fluency throughout. Any sections that seem overly general or lack academic rigour should be rewritten. This will raise the manuscript's general level of excellence and professionalism.
8) More references should be cited to back up the statement regarding the current understanding of the connection between circRNA and liver damage. Include current research or important references that discuss how circRNA forms, functions biologically, and contributes to liver ischemia-reperfusion injury. As a result, the claim will be more accurate and valid, giving readers a strong foundation on which to base their own opinions.

---

## Round 0.2 · accepted · Accept

It is a good revision and can be accepted for publication in PeerJ.

·

Basic reporting

This article is using clear and professional english to express Berberine protects hepatocyte from hypoxia/reoxygenation-induced injury through inhibiting circDNTTIP2.This article's literature references is sufficient, the background provided is reasoned and have professional article structure, figures, tables to support the hypothesis.

Experimental design

The experimental design is reasonable, the method is scientific and the demonstration is sufficient,the methods described with sufficient detail and information can be replicated.

Validity of the findings

The topic selection of the paper is novel, the experimental results can be replicated, and the basic principle and benefits of the paper are clearly explained. The basic data provided by the experiment are robust, statistically reliable and controllable. The conclusions are well stated and strongly related to the original research questions.

Additional comments

Zhu et al. investigated the role of berberine, a common quaternary ammonium alkaloid, in hepatic ischemia-reperfusion injury.  Berberine has a protective effect on H/R-induced hepatocyte damage by inhibiting a novel circRNA, circDNTTIP2, resulting in reduction of caspase 3-related cell apoptosis and the release of inflammatory cytokines,this article has novel topic selection, clear description, sufficient experimental data, smooth expression standard, and credible conclusion.

Reviewer 2 ·

Basic reporting

no comment

Experimental design

no comment

Validity of the findings

no comment

Additional comments

no comment

Reviewer 3 ·

Basic reporting

The authors have addressed all my comments.

Experimental design

no comment

Validity of the findings

no comment